# Multiplexed Proteomic Approach for Identification of Serum Biomarkers in Hepatocellular Carcinoma Patients with Normal AFP

**DOI:** 10.3390/jcm9020323

**Published:** 2020-01-23

**Authors:** Young-Sun Lee, Eunjung Ko, Eileen L. Yoon, Young Kul Jung, Ji Hoon Kim, Yeon Seok Seo, Hyung Joon Yim, Kyun-Hwan Kim, So Young Kwon, Jong Eun Yeon, Soon Ho Um, Kwan Soo Byun

**Affiliations:** 1Department of Internal Medicine, Korea University College of Medicine, Seoul 02841, Korea; lys810@korea.ac.kr (Y.-S.L.); eunjungk@hanmail.net (E.K.); 93cool@hanmail.net (Y.K.J.); kjhhepar@naver.com (J.H.K.); gandorie@gmail.com (Y.S.S.); gudwns21@korea.ac.kr (H.J.Y.); umsh@korea.ac.kr (S.H.U.); kwsbyun@unitel.co.kr (K.S.B.); 2Department of Internal Medicine, Inje University College of Medicine, Seoul 01757, Korea; mseileen80@naver.com; 3Department of Pharmacology, Konkuk University School of Medicine, Seoul 05029, Korea; khkim10@kku.ac.kr; 4Department of Internal Medicine, Konkuk University School of Medicine, Seoul 05029, Korea; sykwonmd@kuh.ac.kr

**Keywords:** mass spectrometry, multiple reaction monitoring, serum biomarker, hepatocellular carcinoma, Trim22, alpha fetoprotein

## Abstract

Alpha fetoprotein (AFP) has been used as a serologic indicator of hepatocellular carcinoma (HCC). We aimed to identify an HCC-specific serum biomarker for diagnosis using a multiplexed proteomic technique in HCC patients with normal AFP levels. A total of 152 patients were included from Guro Hospital, Korea University. Among 267 identified proteins, 28 and 86 proteins showed at least a two-fold elevation or reduction in expression, respectively. Multiple reaction monitoring (MRM) analysis of 41 proteins revealed 10 proteins were differentially expressed in patients with liver cirrhosis and HCC patients with normal AFP. A combination of tripartite motif22 (Trim22), seprase, and bone morphogenetic protein1 had an area under receiver operating characteristic of 0.957 for HCC diagnosis. Real-time PCR and western blot analysis of the paired tumor/non-tumor liver tissue in HCC revealed a reduced expression of Trim22 in the tumor tissue. Also, serum levels of Trim22 were significantly reduced in HCC patients with normal AFP compared to those with liver cirrhosis (*p* = 0.032). Inhibition of Trim22 increased cellular proliferation in human hepatoma cell lines, whereas overexpression of Trim22 decreased cellular proliferation in hepatoma cell lines. In conclusion, the combination of three serum markers improved the chance of diagnosing HCC. MRM-based quantification of the serum protein in patients with normal AFP provides the potential for early diagnosis of HCC.

## 1. Introduction

The high mortality associated with hepatocellular carcinoma (HCC) is partly caused by the diagnosis of HCC at an advanced stage that cannot be treated effectively. Alpha-fetoprotein (AFP), as a serologic indicator of HCC, is often elevated in non-cancerous conditions and is not elevated in as many as 40% of HCC cases [1,2]. In patients with a normal range of AFP, other proteins, such as protein induced by vitamin K absence or antagonist II (PIVKA II) [1,2,3] or AFP L3, may have an additional diagnostic role [4]. However, in many cases of HCC, the levels of these markers are elevated at later disease stages. Due to the limited sensitivity compared to radiological imaging, serum AFP as a tool for the surveillance and diagnosis of HCC has been removed from recent HCC guidelines in many continents [5,6,7]. The lack of a simple and reliable quantitative blood test for surveillance and/or diagnosis of HCC has hindered the development of a noninvasive diagnosis of HCC.

In addition to the limited numbers of target proteins that are available for HCC diagnosis and management, there are technical issues that need to be overcome. Immuno-blotting techniques, which are one of the most common methods used for quantitative measurements of aberrant proteins, are expensive and antibody development is labor intensive.

Recent advances in mass spectrometry (MS)-based protein analysis [8] have allowed the discovery and validation of potential biomarkers from clinical samples in a high-throughput manner [9,10,11]. MS involves the ionization of samples, separation of the proteins, and determination of their relative molecular weights. In addition to results from tissue sample analysis, proteomic analysis of serum samples from solid organ cancers, including gastric and colon, have been reported recently [12]. Blood samples are readily accessible and have obvious advantages over those obtained via invasive procedures, such as tissue biopsy specimens. In patients with gastric cancer [13], six unique serum proteomic signatures with mass/charge values of 5098, 8592, 8610, 11,468, 11,804, and 50,140 were identified. In colorectal cancer patients [14], a blood-based proteomic study revealed an elevation of serum S100A9, Apo A1, C9, protein 14-3-3b, ALDH1, and HSP27, and demonstrated that elevation of these factors was diagnostic, prognostic, and could be used to predict therapeutic response.

Multiple reaction monitoring (MRM) MS [8] is a multiplexed targeted proteomic platform that can facilitate the discovery and validation of potential biomarkers from large numbers of clinical samples in a high-throughput manner.

The aim of our study was to analyze serum proteins and to identify potential HCC-specific serum biomarkers for the early detection of HCC using multiplexed proteomic techniques, such as nano-LC/MS/MS and MRM, in HCC patients with normal AFP levels.

## 2. Methods

### 2.1. Patients and Diagnosis

Sera were collected from HCC patients with normal AFP (HCC group) and patients with liver cirrhosis as controls (control group). Diagnosis of HCC was confirmed as per international guidelines based on liver dynamic images with computed tomography scans and magnetic resonance imaging or liver biopsy [5,6]. The protocol was approved by the Institutional Review Board of Guro Hospital, Korea University Medical College (KUGH11188-002). Written informed consent was obtained from each patient prior to enrollment in accordance with the ethics committee’s recommendation. Guidelines for our institution and the study protocol conformed to the ethical guidelines of the 1975 Declaration of Helsinki as reflected in a priori approval by the institution’s human research committee.

### 2.2. Enzymatic In-Gel Digestion and Nano-LC-ESI-MS/MS Analysis

Differentially expressed proteins in HCC patients were identified through intensive profiling of serum proteomes with nano-LC/MS/MS using pooled sera from liver cirrhosis cases (*n =* 5) and HCC with normal AFP cases (*n =* 5). Proteins separated by SDS-PAGE were excised from gels and destained with 50% acetonitrile (ACN) containing 50 mM NH_4_HCO_3_. These gel pieces were then dehydrated and digested. Following digestion, tryptic peptides were extracted and dried, and concentrated in 0.1% formic acid using C18 ZipTips (Millipore, MA, USA) before MS analysis. Tryptic peptides were loaded onto a fused silica microcapillary column (12 cm × 75 µm) packed with C18 reversed phase resin (5 µm, 200 Å). Liquid chromatography (LC) separation was conducted and the column was directly connected to an LTQ linear ion-trap mass spectrometer (Finnigan, CA, USA) equipped with a nano-electrospray ion source. All spectra were acquired in the data-dependent scan mode. Each full MS scan was followed by five MS/MS scans corresponding to the most intense to the fifth most intense peaks of the full MS scan.

### 2.3. Database Searching

We identified the acquired LC-ESI-MS/MS fragment spectra in BioWorksBrowser^TM^ (version Rev. 3.3.1 SP1, Thermo Fisher Scientific Inc., Waltham, MA, USA) by searching the National Center for Biotechnology Information (http://www.ncbi.nlm.nih.gov/) non-redundant human database, including the reverse database with the SEQUEST search engine. We performed experiments in triplicate and selected proteins identified more than twice in the triplicate experiments. We used a label-free protein quantification method as previously described [15]; a protein was said to be specific to a certain sample when its quantity was at more than twice that present in another sample.

### 2.4. Target Protein Selection and LC-MRM/MS Assay

A total of 267 proteins were identified and quantified in the sera of HCC patients, of which 28 proteins and 86 proteins showed an increase or decrease at least two-fold, respectively. Among these, 41 proteins that showed consistent trends in three replicative analyses were selected for MRM analysis. For MRM analysis, samples (each 50 µg) from each group were resolved, denatured, and reduced. Alkylated HCC samples were subjected to in-solution digestion with sequencing grade modified trypsin (Promega, Madison, WI, USA) overnight at 37 °C. MRM was performed on a QTRAP 5500 hybrid triple quadrupole/linear ion trap mass spectrometer (Applied Biosystems, Foster City, CA, USA) equipped with a nanospray ionization source for quantitative analysis of specific peptides of proteins of interest. A given MRM Q1/Q3 ion value (precursor/fragment ion pair) was monitored to select a specifically targeted peptide corresponding to the candidate proteins. MRM measurement was obtained in triplicate for each target peptide.

### 2.5. mRNA and Protein Expression of Trim22 in Tumor/Non-Tumor Tissues of HCC Patients

Different sets of tumor and non-tumor background liver tissue were obtained after resection of 21 HCC patients to examine mRNA expression of tripartite motif22 (Trim22). For real-time PCR quantification, total RNA was extracted with Trizol (Gibco BRL, Gaithersburg, MD, USA), according to the manufacturer’s instructions. Reverse transcription was performed with a cDNA synthesis kit (Roche, Basel, Switzerland). PCR amplifications were performed in a 25-μL reaction mix containing 300 nM of the forward and reverse primers, and SYBR Green I PCR reagents (Applied Biosystems, Foster City, CA, USA). Amplified signals were detected continuously by real-time quantitative PCR on an ABI Prism 7000 Sequence Detection System (Applied Biosystems, Foster City, CA, USA). The following real-time PCR amplification protocol was used: (i) Initial denaturation at 95 °C for 10 min; and (ii) three-segment amplification and quantification comprising 40 cycles of 95 °C for 60 s, 60 °C for 30 s, and 72 °C for 30 s. Primers used for PCR were as follows: Forward 5′-tctgagtgggatgctgcaag, reverse 5′-gccgaagacaccaaaagcag.

### 2.6. Western Blot Analysis of Trim22 Using Tumor/Non-Tumor Liver Tissues from HCC Patients and Other Malignancy Patients with Liver Metastasis

Tumor and non-tumor background liver tissue from 19 HCC patients were obtained after resection. Furthermore, tumor/background non-tumor liver tissues from non-HCC patients with liver metastasis were obtained for comparison: Colorectal cancer with liver metastasis (*n =* 7), bile duct cancer (*n =* 2), breast cancer (*n =* 1), neuroendocrine tumor (*n =* 1), and ampulla of Vater cancer (*n =* 1). Protein was extracted with RIPA buffer (50 nM Tris-HCl (pH 7.5), 1% NP-40, 0.25% Na-deoxycholate, 150 mM NaCl, 1 mM EDTA, 2 mM EGTA) containing protease and phosphatase inhibitors (1 mM phenylmethylsulfonyl fluoride, 0.1 mM *N*-tosyl-l-phenylalanine chloromethyl ketone, 1 μg/mL aprotinin, 1 μg/mL pepstatin, 0.5 μg/mL leupeptin, 1 mM NaF, 1 mM Na_4_P_2_O_4_, 2 mM Na_3_VO_4_). Protein concentration was determined using a Bio-Rad protein assay (Bio-Rad Laboratories Inc., Melville, NY, USA). A total of 50 μg of protein was used for the immuno-blotting analysis. Nonspecific binding sites were blocked with SuperBlock Blocking Buffer (Pierce, Rockford, IL, USA). Membranes were incubated with primary antibody directed against Trim22 (Sigma, St. Louis, MO, USA), overnight at 4 °C with gentle agitation. Immunoreactivity was detected with horseradish-peroxidase-conjugated secondary antibodies and enhanced chemiluminescence reagents (MEN Life Science, Boston, MA, USA), and then quantified by Scion Image analysis.

### 2.7. Western Blot Analysis of Serum Trim22 Expression in Hepatocellular Carcinoma Patients with Normal AFP

Serum samples from liver cirrhosis (*n =* 30) and HCC patients with normal AFP (*n =* 39) were extracted for serum Trim22 expression analysis. After albumin depletion, serum protein was extracted as described previously. A total of 50 μg of protein was used for the immunoblotting analysis and membranes were incubated with primary antibody directed against Trim22 (Sigma, USA), overnight at 4 °C with gentle agitation.

### 2.8. Statistical Analysis

Categorical data were compared using the chi-square or Fisher’s exact test as indicated. Continuous variables were compared using the Students *t* test or Mann–Whitney test for variables with a non-parametric distribution. Receiver operating characteristics (ROC) curve analysis was used to determine the optimal cutoffs of continuous variables by choosing the point along the curve that maximized the sum of sensitivity and specificity. Significance was defined by a *p* value of <0.05. This analysis was carried out using SPSS (version 24.0, Chicago, IL, USA).

## 3. Results

### 3.1. Baseline Characteristics

Sera were collected from HCC patients with normal AFP and patients with liver cirrhosis as controls. Baseline characteristics of each group, including biochemical liver function, degree of inflammation, synthetic liver function, and causes of liver disease, were not significantly different (Table 1). Mean serum AFP levels were 4.04 and 4.25 ng/mL in group 1 and 2, respectively (*p* = 0.800). More than half of the patients with HCC had tumors less than 5 cm in diameter and most tumors were single lesions (71.4%) (Table 1).

### 3.2. MRM/MALDI-TOF MS/MS Analysis of Serum Proteins

Serum samples from patients with liver cirrhosis in the current study were selected as controls for comparison with HCC patients with normal AFP levels. A total of 267 proteins were identified and quantified in the sera of HCC patients, of which 28 and 86 proteins showed two-fold increased or decreased expression relative to their expression in liver cirrhosis samples, respectively. Among the 114 proteins that showed altered expression in HCC patients with normal AFP levels, 41 proteins that consistently showed reliable trends across three replicative analyses were selected for MRM analysis. Among those 41 proteins, MRM/MALDI-TOF MS/MS revealed that 10 proteins, namely glypican3 (GPC3), squamous cell carcinoma antigen (SCCA), haptoglobin, C3 precursor (C3), seprase, hemoglobin subunit gamma 2 (Hb-γ2), hemoglobin subunit alpha (Hb-α), bone morphogenetic protein 1 (BMP1), teneurin3, and tripartite motif 22 (Trim22), were significantly altered in Group 2 compared to Group 1. Table 2 shows the Q1/Q3 ionic transition and peptide sequences of these 10 proteins.

### 3.3. Averaged Abundance of Target Proteins in HCC Patients with Normal AFP Levels

As shown in Table 3, the averaged abundance of the 10 proteins in the cancer group (Gr.2) was compared to that in the control group (Gr.1). Eight proteins, namely GPC3, SCCA, haptoglobin, C3, seprase, Hb-γ2, Hb-α, and Teneurine3, were elevated in Gr.2 compared to Gr.1. Trim22 and BMP1 had a lower abundance in Gr.2 than Gr.1. The greatest difference in abundance between the HCC group and liver cirrhosis group was observed for hemoglobin subunit alpha, with a factor of 6.25 (cancer/liver cirrhosis, *p* = 0.006). The abundance ratio of cancer/liver cirrhosis of each protein was as follows: GPC3 (1.49, *p* = 0.003), SCCA (1.4, *p* = 0.024), haptoglobin (2.25, *p* = 0.038), C3 (1.38, *p* = 0.005), seprase (1.33, *p* < 0.001), Hb-γ (3.72, *p* = 0.021), Hb-α (6.25, *p* = 0.006), and teneurin3 (2.73, *p* = 0.004), respectively. In addition to BMP1 (0.8, cancer/liver cirrhosis, *p* = 0.013), Trim22 was less abundant in HCC patients with normal AFP than liver cirrhosis patients, and this difference was statistically significant (0.6, cancer/liver cirrhosis, *p* < 0.001).

As shown in Table 3, the sensitivity and specificity of each protein for diagnosing HCC were as follows: GPC3 (61.9%, 100%), SCCA (95.2%, 50%), haptoglobin (85.7%, 60%), C3 (76.2%, 70%) Trim22 (90%, 85.7%), seprase (76.2%, 90%) Hb-γ (66.7%, 80%), Hb-α (66.7%, 100%), BMP1 (95.2%, 71.4%), and teneurin-3 (95.2%, 80%), respectively.

Quantitative data for the 10 proteins were also evaluated in 31 patients using a scatterplot (Figure 1). It is clear from this figure that HCC with normal AFP levels and liver cirrhosis can be differentiated by the concurrent use of multiple HCC biomarker candidates, such as those developed in this study.

### 3.4. Receiver Operating Characteristic (ROC) Curves for Each of the 10 Proteins Were Significantly Altered in HCC Patients with Normal AFP Compared to Those with Liver Cirrhosis

ROC curves for each of these 10 proteins were constructed to investigate the differences between HCC with normal AFP level samples and liver cirrhosis samples, as shown in Table 3 and Figure 2. The area under the curve (AUROC) for each protein was as follows: GPC3 (0.790), SCCA (0.733), haptoglobin (0.791), C3 (0.762), seprase (0.824), Hb-γ (0.721), Hb-α (0.848), BMP1 (0.817), and teneurin3 (0.876). Among those proteins, the AUROC of Trim22 was the highest (0.924).

We further evaluated the ability of different combinations of the three proteins to function as HCC biomarkers by generating ROC curves, as illustrated in Figure 2. The best AUROC, as generated by a combination of the three proteins, was at a value of 0.957, with a sensitivity and specificity of 95.2% and 90.0%, respectively.

### 3.5. mRNA Expression of Trim22 in Tumor/Non-Tumor Background Tissues of HCC Patients

To validate the MRM/MALDI-TOF MS/MS analysis, the most accurate marker, Trim22, was chosen for mRNA expression experiments. Twenty-one different sets of paired HCC tumor/non-tumor background liver tissues were obtained from HCC patients who underwent surgical resection. Six of these 22 patients had elevated serum AFP, with a mean value of 9023.7 ng/mL (range, 228–45,859 ng/mL). The relative ratio of tumor/non-tumor mRNA expression of Trim22 was less than 1.0 in 14 of 21 patients (66.6%) and most of them had Trim22 levels less than 0.5 (11 of 14, 78.5%). Even in the six HCC patients with elevated AFP levels, Trim22 mRNA analysis revealed decreased expression in the tumor compared to the background non-tumor in five of the six patients (83.3%) and most of these patients (4/5) had Trim22 mRNA levels less than 0.5.

### 3.6. A Western Blot Analysis of Trim22 Expression in Tumor/Non-Tumor Liver Tissues in Hepatocellular Carcinoma and Non-Hepatocellular Carcinoma Patients with Liver Metastasis

Because immuno-blotting for certain proteins is the method most frequently used in clinical situations, we examined liver tissue samples from another set of 19 HCC patients and 12 non-HCC patients with liver metastasis to evaluate Trim22 protein expression using antibodies. In patients with HCC, the average ratio of Trim22 expression in tumor/non-tumor liver tissue was 0.72 (*p* = 0.008). In non-HCC with liver metastasis patients, the ratio of Trim22 expression in tumor/non-tumor liver tissue was 1.06, indicating no difference in the expression of this protein between tumor and non-tumor tissues (Figure 3).

### 3.7. Serum Trim22 Expression in Hepatocellular Carcinoma Patients with Normal AFP

We also examined serum Trim22 from 30 LC patients and 39 HCC patients for validation of these results. The characteristics are summarized in Table 4. Consistent with the MRM finding, serum Trim22 expression was significantly reduced in HCC patients compared with LC patients as shown in Figure 4 (*p* = 0.034).

### 3.8. The Regulation of Hepatocytes Proliferation by Trim22

Next, we investigated hepatocyte proliferation after inhibition or overexpression of Trim22 expression in HepG2 and SNU 475 cells (Figure 5a–d). When Trim22 expression was inhibited, proliferation of HepG2 and SNU475 was significantly increased compared to the negative control (Figure 5e,f). When Trim22 was overexpressed, proliferation of HepG2 and SNU475 was slightly decreased compared to the negative control (Figure 5g,h).

## 4. Discussion

Recently, the role of serum AFP as a surveillance tool or diagnostic marker for HCC has been reduced due to its limited sensitivity and specificity. AFP is often elevated in non-cancerous conditions, such as severe inflammation, and is not elevated in as many as 40% of HCC patients [1,2,16,17]. Other proteins, such as PIVKA II or AFP L3, may aid in diagnosis [3,4], but in many cases of HCC, the expression of these biomarkers is only elevated in later stages. We aimed to discover possible serologic indicators of HCC using mass spectrometry-based multiple reaction monitoring (MRM) in HCC patients with a normal AFP level.

After extensive profiling of protein expression in serum samples from HCC patients with normal AFP levels, as well as liver cirrhosis patients for a control, we identified 10 proteins that were significantly altered in HCC patients compared to controls among the 41 MRM-targeted proteins. Of these, eight proteins, glypican 3 (GPC3), squamous cell carcinoma antigen (SCCA), haptoglobin, C3 precursor, seprase, hemoglobin subunit gamma, hemoglobin subunit alpha, and teneurin-3, showed higher relative peaks in HCC cases with normal AFP levels than the liver cirrhosis controls. Trim22 and BMP1 expression was reduced in HCC cases with normal AFP levels.

Previous studies that used an antibody-based immune blotting method reported that GPC3 [18,19] and SCCA [20,21] were elevated in 50% and 70% of HCC patients, respectively, and suggested that these proteins were useful tools for early diagnosis of HCC. In patients with normal AFP, little is known about the diagnostic significance of these markers. AFP-L3 [4] and Golgi protein 73 [22] have been reported to be elevated in 40% and 67% of HCC patients with normal AFP levels, respectively. In our HCC patients with normal AFP, the sensitivity and specificity of most of the 10 proteins we identified was more than 60%.

In our study, the most significantly altered protein in HCC patients with normal AFP levels was Trim22, with a sensitivity of 90% and specificity of 85.7%. Trim (tripartite motif containing) proteins are E3 ubiquitin ligases that are involved in oncogenic processes, such as transcriptional regulation, cell proliferation, and apoptosis [23]. Trim19, which is encoded by the promyelocytic leukemia gene, is involved in t(15; 17) translocation [23]. Trim22 is known to be expressed in several human tissues, is highly upregulated in response to type I and type II interferon, and has been shown to restrict the replication of a number of viruses, including encephalomyocarditis virus, hepatitis B virus, and human immunodeficiency virus type I [24,25]. Moreover, Trim22 may have a role in cellular differentiation, proliferation, and carcinogenesis linked to p53 [24]. In HCC, Trim22 expression was decreased during early intrahepatic recurrence [26]. Our serum proteomic analysis revealed that Trim22 in patients with normal AFP levels was reduced to 60% of the expression levels in patients with liver cirrhosis. Although biomarkers usually have elevated levels in association with a particular disease state, the lower peak intensity of Trim22 in HCC samples was very significant. The AUROC of Trim22 for diagnosing HCC was 0.924. To confirm the MRM result, we performed western blot analysis using a different set of patients’ sera. As a result, we identified that serum Trim22 was significantly reduced in HCC patients compared to liver cirrhosis controls. Most HCC and liver cirrhosis control patients in our study were infected with the hepatitis virus and this may have affected the protein expression of Trim22. However, in our study, most patients with liver cirrhosis were also infected with hepatitis B virus (HBV). Further studies are needed to define the role of Trim22 in the carcinogenesis of HCC.

In addition to reduced serum Trim22 expression, Trim22 protein expression was reduced to 72% in liver tumor tissue compared to the background non-tumor tissue in HCC patients. Using the different sets of tumor and non-tumor tissues from non-HCC patients with liver metastasis not infected with HBV, we showed by western blotting analysis that Trim22 protein expression in tumor tissue was not different from that in the non-tumor tissue background. Further studies are needed to define the role of Trim22 in the carcinogenesis of HCC. In addition to liver tissue protein expression, the relative expression of Trim22 mRNA in tumor tissue compared to non-tumor tissue was also reduced in most of the HCC cases we examined. Inhibition of Trim22 increased cellular proliferation in human hepatoma cell lines, whereas overexpression of Trim22 decreased cellular proliferation in hepatoma cell lines. Because Trim22 is related to interferon signaling, there is a possibility that cell proliferation was influenced by this signaling by HBV in hepatoma cell lines [27,28]. In our study, however, cell proliferation by Trim22 regulation showed a similar pattern in HepG2 that does not contain HBV and SNU 475 that is infected with HBV. Although the expression of Trim22 is influenced by its association with a particular disease state, which includes HBV infections, the lower peak intensity of Trim22 in the HCC samples was significant.

The other proteins we identified in our screen also had relatively high AUROC values for the diagnosis of HCC. Even in HCC patients with elevated serum AFP and PIVKA II levels, the AUC of these two proteins was reported as only 0.79 and 0.82, respectively. Our candidate proteins showed equal to or higher power for diagnosing HCC with normal AFP levels than AFP and PIVKA II. Therefore, these proteins may be useful surveillance markers in patients with normal AFP and PIVKA II, especially in patients that receive HCC surveillance after curative resection.

In our study, haptoglobin, hemoglobin gamma, hemoglobin alpha, and teneurin3 had the highest averaged ratio compared to the liver cirrhosis control. Sarvari et al. [8] reported that expression of the haptoglobin a-2 isoform was significantly increased in HCC patients compared to liver cirrhosis controls. Haptoglobin is produced by malignant ovarian epithelium, renal cell carcinoma, and HCC cells. Our results are therefore consistent with other studies that reported an alteration of haptoglobin expression, suggesting that haptoglobin [8,29] or specific haptoglobin glycoform alterations [30] may have an important role in HCC.

Seprase, also known as fibroblast activation protein (FAP), is an integral membrane serine peptidase [31]. Proteolytic activity of seprase has been shown to induce cell invasion of the extracellular matrix (ECM) and also to support tumor growth and proliferation. In our study, serum seprase showed a higher peak intensity in HCC samples than liver cirrhosis controls. In patients with non-small cell lung cancer [32] and colorectal cancer [33], elevated FAP expression was reported to be related with tumor progression and poor survival.

Bone morphogenetic proteins (BMPs) belong to the TGF-B superfamily; proteins in this superfamily are involved in homeostasis of diverse tissue and organs by regulating cellular differentiation, proliferation, apoptosis, and motility. In HCC, BMP 4, 7, and 9 expression is associated with poor prognosis, the epithelial to mesenchymal transition, and FGF2 expression [34,35]. In our study, BMP1 showed a lower peak intensity in HCC patients with normal AFP levels than liver cirrhosis patients. Further studies are required to determine the role of BMP1 in HCC.

Among the 41 target proteins, we included a previously reported biomarker that showed altered expression in HCC, such as Golgi protein73 [22], osteopontin [36], HSP27 [37], alpha2 HS, preapoproteins, amyloid A, and proteasome [38]. However, serum levels of these proteins were not significantly different between HCC patients with normal AFP and liver cirrhosis controls. Feng et al. reported detection of heat shock protein 27 in the serum of 90% of HCC patients that they evaluated [37]. However, in their study, most HCC patients had an elevated serum AFP level, large tumor size, daughter nodules, and more advanced stage tumor than our study patients. These finding may suggest that previously identified biomarkers, such as HSP27, were not altered in our study.

In our study, the 41 proteins identified by serum proteomics profiling of HCC samples and liver cirrhosis controls were simultaneously monitored using MRM. MRM-based MS has been widely used to assay the abundance of newly identified serologic proteins because of its high sensitivity in mass detection and high specificity for target peptides. Previous studies targeting HCC serum markers have commonly selected only two or three target proteins and then analyzed the sensitivity and specificity using different cut-off levels [1,2,3,4]. In contrast to these previous studies, we simultaneously monitored 41 proteins and identified 10 proteins that showed altered expression in HCC patients with normal AFP. Even a single marker, Trim22, alone showed high power for diagnosing HCC. The combination of Trim22, BMP1, and seprase had an AUROC of 0.957 for the diagnosis of HCC. We selected Trim22 for immunoblotting analysis using patients’ sera and confirmed lower expression in HCC patients with normal AFP than liver cirrhosis controls. Protein and mRNA analysis using different sets of tumor/non-tumor background liver tissue samples from HCC patients and non-HCC patients with liver metastasis confirmed the lower expression in tumor tissue than in non-tumor background liver tissue in HCC patients. Also, consistent with MRM findings, we identified that serum Trim22 was significantly reduced in HCC patients with normal AFP compared with liver cirrhosis controls.

There are some limitations in our study. First, most patients involved in this study were infected HBV. Therefore, the biomarkers for the diagnosis of HCC may be different in other etiologies of HCC, such as hepatitis C virus infection, alcoholic liver disease, and nonalcoholic liver disease. This limitation induced an imbalance of the etiology of HCC between the liver cirrhosis group and the HCC group. In order to apply this biomarker for diagnosis of HCC in real clinical practice, further validation studies are required that include HCC patients with various etiologies of other chronic liver diseases. Another limitation of our study is the small sample size because MRM is still a labor-intensive procedure. However, in our study, we triplicated our mass spectrometry-based quantification results in each sample with or without albumin depletion after profiling serum proteins. We also used one to five peptides to identify each target protein. Third, the meaning of our results is limited without validation. Although we conducted an additional validation analysis by measuring serum Trim22 expression to overcome this limitation, further validation study is required. Fourth, only four patients (19%) had small-sized (≤3 cm) HCC in the MRM analysis. Although the validation analysis for serum Trim22 included 11 patients with small-sized HCC, further studies are required that emphasize patients with small-sized HCC. Finally, we only included HCC patients only with normal AFP. Although the purpose of this study is to identify potential HCC-specific serum biomarkers in HCC patients with normal AFP levels, further studies that include HCC patients at high AFP levels would be both valuable and necessary.

## 5. Conclusions

We used a multiplexed proteomic approach to identify proteins associated with HCC. The combination of the three serum markers of Trim22, seprase, and BMP-1 could diagnose HCC with good sensitivity and specificity. A mass spectrometry-based multiplexed quantification of serum proteins in patients with normal AFP levels identified several potential candidates for surveillance of HCC.

## Figures and Tables

**Figure 1 jcm-09-00323-f001:**
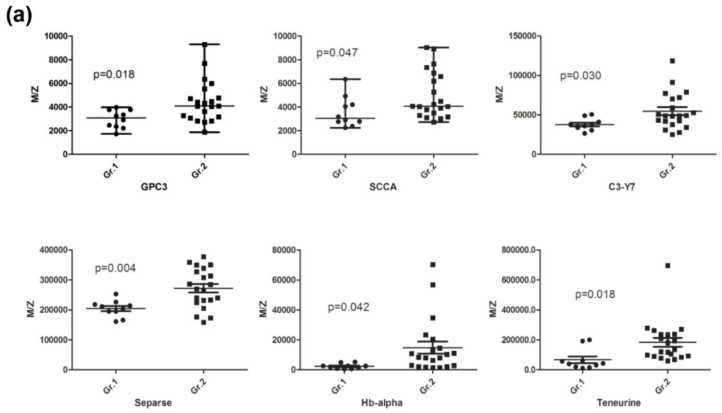
A comparison of abundance for target proteins was conducted between control (LC, Gr.1) and HCC with normal AFP (Gr.2). Scatterplot of eight proteins with higher (**a**) and two proteins with lower (**b**) peak intensity in HCC with normal AFP compared with liver cirrhosis are shown. Plots show the median (horizontal bar) and inter-quartile ranges, and the bars show the minimum and maximum values.

**Figure 2 jcm-09-00323-f002:**
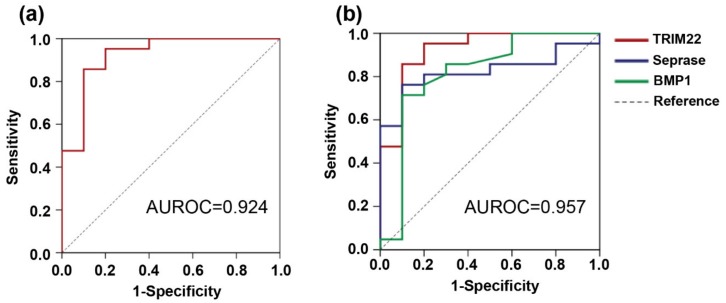
AUROC for Trim22 alone (**a**) and for the combined three proteins, including seprase, BMP1, and Trim22 (**b**), is shown. The ability of each biomarker candidate to discriminate HCC from cirrhosis was evaluated.

**Figure 3 jcm-09-00323-f003:**
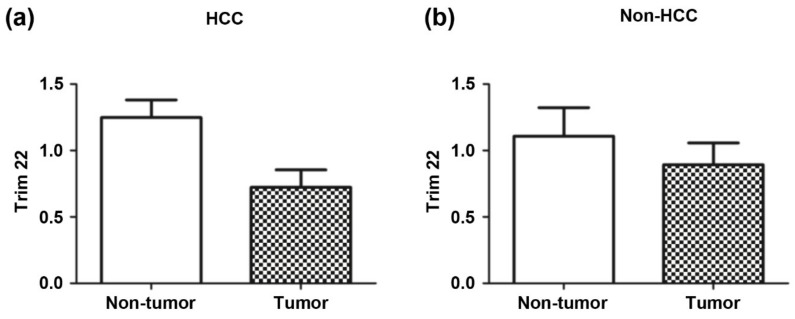
Tim22 protein expression in HCC patients with normal AFP (**a**) are compared with non-HCC patients with liver metastasis (**b**). Trim22 expressions are expressed as the relative ratio of protein expression in tumor compared to non-tumor background liver.

**Figure 4 jcm-09-00323-f004:**
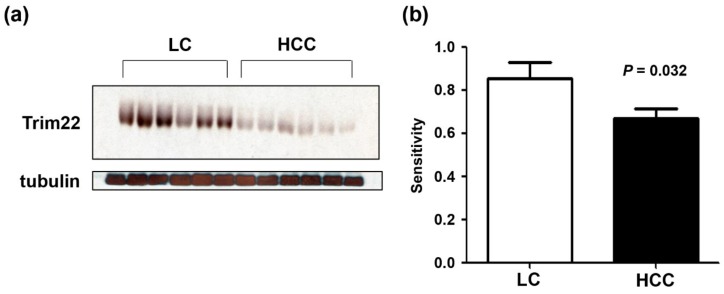
Serum Trim22 was compared in liver cirrhosis (*n =* 30) and HCC patients with normal AFP (*n =* 39). Representative western blot analysis is shown (**a**). Compared with liver cirrhosis controls, relative expressions of serum Trim22 normalized with tubulin was significantly reduced in HCC patients (**b**).

**Figure 5 jcm-09-00323-f005:**
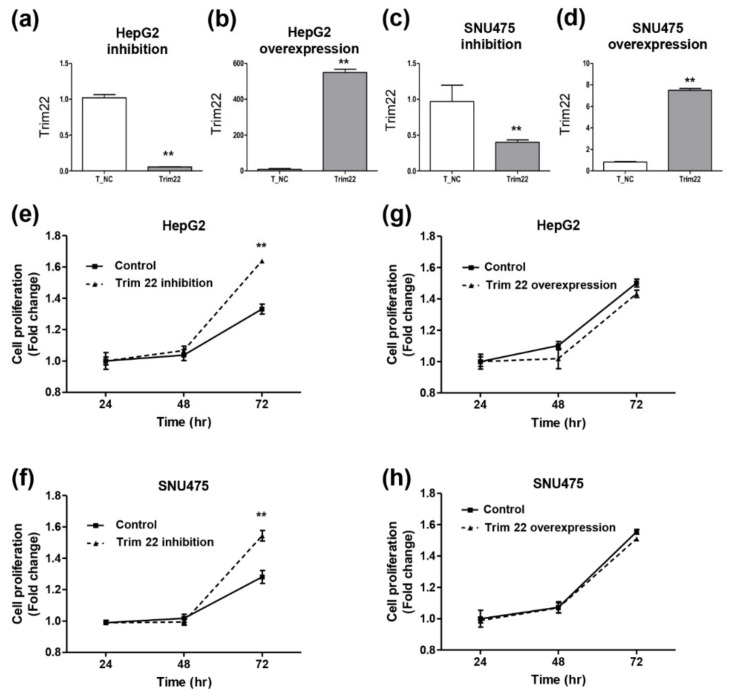
Trim22 is effectively inhibited (**a**) and upregulated (**b**) in HepG2 cells. Trim22 is effectively inhibited (**c**) and upregulated (**d**) in SNU475 cells. Cell proliferation was analyzed after inhibition of Trim22 in HepG2 (**e**) and SNU475 (**f**). Cell proliferation was analyzed after overexpression of Trim22 in HepG2 (**g**) and SNU475 (**h**). ** indicates *p*-value < 0.01.

**Table 1 jcm-09-00323-t001:** Baseline characteristics of patients with liver cirrhosis and HCC patients with normal AFP for multiple reaction monitoring.

	LC Patients (Group 1, *n =* 10)	HCC Patients with Normal AFP (Group 2, *n =* 21)	*p*-Value
Age, years	51.2 ± 10.2	62.8 ± 9.9	0.014
Sex (M/F)	7/3	16/5	0.415
Etiology HBV/HCV/Alcohol	10/0/0	14/2/5	0.116
Child-Pugh score A/B/C	9/1/0	20/1/0	0.579
Total bilirubin (mg/dL)	0.90 ± 0.35	0.75 ± 0.39	0.303
Albumin (g/dL)	4.23 ± 0.42	3.90 ± 0.50	0.096
PT (INR)	1.07 ± 0.06	1.08 ± 0.06	0.762
AFP (ng/mL)	4.04 ± 2.16	4.25 ± 2.25	0.800
Tumor number 1/2~3/4 (*n*, %)	NA	15/6/0 (71.4/19.1/9.5, %)	NA
Tumor size <3, 3~5, >5 cm, (*n*, %)	NA	4/8/9 (19/38.1/42.9, %)	NA
Tumor differentiation 1/2/3/4, (*n*, %)	NA	3/10/6/0 (15.8/52.6/31.6/0, %)	NA
BCLC, 0/A/B/C/D, (*n*, %)	NA	1/11/8/1 (4.8/52.4/38.1/4.8, %)	NA

Values are represented as mean ± SD, LC, Liver cirrhosis; HCC, Hepatocellular carcinoma; HBV, Hepatitis B virus; HCV, Hepatitis C virus; PT, Prothrombin time; AFP, Alpha-fetoprotein; BCLC, Barcelona clinic liver cancer; NA, not applicable.

**Table 2 jcm-09-00323-t002:** Peptide sequence and Q1/Q3 (precursor/fragment ion pair) ionic transition of 10 serum proteins that showed altered expression in Hepatocellular carcinoma patients with normal AFP compared with liver cirrhosis controls.

NCBI GI	Protein	Peptide Sequence	Q1	Q3
23271174	Glypican 3 (GPC23)	VFGNFPK	404.72	562.3
239552	squamous cell carcinoma antigen (SCCA)	VLHFDQVTENTTGK GQWEK	794.9 324.16	977.49 462.23
386783	Haptoglobin	TEGDGVYTLNDK	656.31	1081.52
115298678	complement C3 (C3)	GYTQQLAFR	542.28 542.28	762.43 863.47
4504345	Hemoglobin subunit gamma 2 (Hb-γ2)	LLVVYPWTQR	637.87 637.87	687.36 850.42
6715607	Hemoglobin subunit alpha (Hb-α)	VGAHAGEYGAEALER	765.37	1094.51
122937400	Teneurin-3	SDETGWTTFFDYDSEGR	671.61	726.31
16933540	Seprase	TQEHIEESR	564.77	770.38
148745745	Bone morphogenetic protein 1 (BMP-1)	DGFWR.2/y3	340.66	508.27
116283348	Tripartite motif-containing antigen 22 (Trim22)	HLANIVER	476.27	701.39

**Table 3 jcm-09-00323-t003:** The averaged abundance of 10 target proteins in an MRM-ased analysis of serum samples from hepatocellular carcinoma with normal AFP levels and liver cirrhosis patients. Sensitivity, specificity, and area under the receiver operating curve (ROC) values of each protein are shown.

Protein	Averaged Abundance (fmol)		Ratio	*p*-Value	Sensitivity/Specificity	AUC	95% CI
	HCC	LC	(HCC/LC)				
GPC3	4438 ± 1763	2978 ± 766	1.49	0.003	61.9/100	0.790	0.632–0.949
SCCA	10,004 ± 2989	7914 ± 165	1.26	0.019	57.1/90.0	0.733	0.555–0.912
	5012 ± 1986	3578 ± 1299	1.40	0.024	95.2/50.0	0.733	0.537–0.929
Haptoglobin	18,339 ± 18,773	8164 ± 7024	2.25	0.038	85.7/60.0	0.791	0.521–0.917
C3	410,703 ± 16,297	31,873 ± 5311	1.29	0.027	76.2/70.0	0.679	0.488–0.869
	54,830 ± 22,951	37,815 ± 7462	1.38	0.005	66.7/80.0	0.762	0.590–0.934
Hb-γ2	535,196 ± 720,000	144,343 ± 93,717	3.71	0.024	66.7/80.0	0.721	0.540–0.902
	303,225 ± 390,000	81,560 ± 54,269	3.72	0.021	66.7/80.0	0.714	0.532–0.897
Hb-α	14,830 ± 18,335	2372 ± 1635	6.25	0.006	66.7/100	0.848	0.713–0.982
Teneurin-3	183,688 ± 138,000	67,378 ± 69,984	2.73	0.004	95.2/80.0	0.876	0.000–1
Seprase	272,230 ± 65,326	204,450 ± 27,254	1.33	<0.001	76.2/90.0	0.824	0.676–0.972
BMP-1	288,682 ± 47,235	361,200 ± 72,619	0.80	0.013	90.0/71.4	0.817	0.630–1.003
Trim22	1,190,000 ± 240,000	1,950,000 ± 420,000	0.61	<0.001	90.0/85.7	0.924	0.813–1

GPC 23, Glypican 3; SCCA, Squamous cell carcinoma antigen; Hb, Hemoglobin; BMP-1, Bone morphogenetic protein 1; Trim22, Tripartite motif-containing.

**Table 4 jcm-09-00323-t004:** Baseline characteristics of liver cirrhosis patients and HCC patients with normal AFP for serum Trim22 expression.

	LC Patients (*n =* 30)	HCC Patients with Normal AFP (*n =* 39)	*p*-Value
Age, years	58.4 ± 9.67	67.1 ± 11.4	0.001
Sex (M/F)	16/14	32/7	0.010
Etiology HBV/HCV/Alcohol	22/0/8/0	23/3/13	0.213
Child-Pugh score A/B/C	29/1/0	37/2/0	0.717
Total bilirubin (mg/dL)	0.81 ± 0.36	0.72 ± 0.31	0.267
Albumin (g/dL)	4.25 ± 0.37	3.97 ± 0.46	0.008
PT (INR)	1.17 ± 0.66	1.06 ± 0.06	0.313
AFP (ng/mL)	4.27 ± 3.81	4.31 ± 2.19	0.956
Tumor number 1/2~3/4 (*n*, %)	NA	30/7/2 (76.9/17.9/5.1, %)	NA
Tumor size <3, 3~5, >5 cm, (*n*, %)	NA	11/15/13 (28.2/38.5/33.3, %)	NA
Tumor differentiation 1/2/3/4, (*n*, %)	NA	8/19/12/0 (20.5/48.7/30.8/0, %)	NA
BCLC, 0/A/B/C/D, (*n*, %)	NA	5/20/9/5/0 (12.8/51.3/23.1/12.8/0.0, %)	NA

Values are represented as mean ± SD, LC, Liver cirrhosis; HCC, Hepatocellular carcinoma; HBV, Hepatitis B virus; HCV, Hepatitis C virus; PT, Prothrombin time; AFP, Alpha-fetoprotein; BCLC, Barcelona clinic liver cancer.; NA, not applicable.

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
