# Peer review of "Multiplexed Proteomic Approach for Identification of Serum Biomarkers in Hepatocellular Carcinoma Patients with Normal AFP"

_jcm, 2020, doi:10.3390/jcm9020323_

Round 1
Reviewer 1 Report
My recommendations are good carry out. The new table No. 4 improve the understanding of patients included. The disussion , especially the last section , is better understable, now.
Author Response
We appreciate this reviewer`s kind comments. We performed additional English proofreading accordance reviewer`s comments that English language and style are fine/minor spell check required. Your insightful comments significantly improved our manuscript.
Reviewer 2 Report
The revision of the manuscript was reviewed. The authors have clarified all comments and improved the manuscript.
Author Response
We appreciate this reviewer`s kind comments. Your insightful comments significantly improved our manuscript.
Reviewer 3 Report
Only 4 patients had small sized HCC in this study, thus I could not understand that the present markers provides potential for early diagnose of HCC.
To conclude as above, the patients with early stage (small size) of HCC should be recruited.
Author Response
Response: We thank the reviewer for the insightful comments and apologize about limitation as reviewer`s comment. Although only 4 patients had small sized HCC in MRM analysis, validation analysis for serum Trim22 included 11 patients with small size HCC. We added further description in discussion. We appreciate reviewer`s valuable comment once again.
Discussion
Fourth, only 4 patients (19%) had small-sized (≤ 3 cm) HCC in MRM analysis. Although validation analysis for serum Trim22 included 11 patients with small size HCC, further studies are required that emphasize patients with small-sized HCC. (Line 394)

This manuscript is a resubmission of an earlier submission. The following is a list of the peer review reports and author responses from that submission.
Round 1
Reviewer 1 Report
Hepatocellular carcinoma ( HCC) is a highly malignant tumour with rising occurrence. In the vast majority of cases, there is an underlying liver cirrhosis that, therefore, is called pre-cancerosis. For this reason, patients with liver cirrhosis should be monitored by of abdominal sonography and effective lab diagnosis in intervals of 3 - 6 months. Sonographic evidence of focus by abdominal sonography is approx. 1 cm. The period of growth of a non-visible nodule (< 1cm) up to 3 cm is approx. 6 months. Sensitivity of abdominal sonography is 65 - 80%, and when using contrast agent sonography at 90 - 95%. <thete ist no realible liver tumour marker yet. Alpha-feto-protein, which is often handled as such, only has a seinsitivity of approx. 60% and a specivity of approx. 90%. In this situation, it is highly topical and necessary to look for other, more effective lab parameters for early recognition of HCC. The three serum markers, Trim 22, Seprase and BMP-1, achieve a good sensitivity and specifity in diagnosis of HCC in combination. How high are these in percent? From the point of view of the clinical hepatologist, it is a limitation that most of of patients involved in the study ( 20 of 31) were infected with the hepatitis-B virus. Considering the heterogenicity of HCC, these promising results cannot be automatically transferred to liver cirrhosis due to hepatitis-C infection, and even less to alcoholic or non-alcoholic fatty liver disease. The documented biomarkers are highly specific proteins with potential structural differences in amino acids in theses different etiological factors. The following must be said, limiting overal positive impression:
There is an imbalance in the two groups: group 1 ( liver cirrhosis n= 10 patients), group 2 ( HCC with normal AFP n=21 patients). Beyond this, there is clearly a greater number of hepatis-B patients in group 1 ( 9 of 10) as compared to group 2 (12 of 21). These share of patients with alcoholic genesis of the chronic liver disease also is very different between the groups: no patient in group 1; 7 patients in group 2 ( 33,3%). Another relevant question is in how many of the 12 patients with hepatitis B HCC developed based chronic hepatitis, i.e. preliminary stage of liver cirrhosis. As has already been found, genesis and stage of the underlying liver disease may be higly relevant in expression of specific proteins. The relevance of the acquired results is, in the end, also reduced by the overall number of patients involved in the study. The applied procedures and methods are at the state of the art and meaningful, but it is not clear if they can also be applied in clinical everyday work. The presentations in the work are designed fact-based and comprehensible, while the discussion is rather long and lacks conciseness.
THe statistical analyses, tables and figures are designed well and comprehensible. The paper was written in an English that is easily comprehensible. All in all, I recommend accepting the paper for publication after revision to resolve the ambguities named.
Reviewer 2 Report
In the present study entitled “Multiplexed Proteomic Approach for Identification of Serum Biomarkers in Hepatocellular Carcinoma Patients with Normal AFP” the authors aimed to analyze serum proteins and to identify potential HCC-specific serum biomarkers for the early detection of HCC using multiplexed proteomic techniques in HCC patients with normal AFP levels. Overall, this is a well-written and interesting work. However, there are several issues that should be addressed.
Comments:
As the authors also mentioned in the limitation, this study includes a small sample size, which can result in low power and unreliable results. Keeping this in mind, only including HCC patients with normal AFP seems to be not appropriate. I would suggest authors to also include HCC patients with high AFP, and perform subgroup analysis for HCC patients with normal AFP. Please give p values in Table 1 and 2. I would suggest authors to describe more about the feasibility of this method in the clinical setting. The authors concluded that “Mass spectrometry based multiplexed quantification of serum proteins in patients with normal AFP levels provided several potential candidates for early diagnosis of HCC”. However, all included patients had already HCC, and it is still unclear, whether identified serum proteins can help to diagnosis in the early phase. Therefore, the conclusion should be revised, and this hypothesis should be evaluated in a large-scale study including patients in the early phase/ suspicion of HCC.
Reviewer 3 Report
The authors identify 10 proteins expressed in HCC patients with normal AFP level by MRM analysis. The combination of Trim22, seprase, and BMP1 have a high AUROC, thus they conclude that these combination markers provide the potential for early diagnosis of HCC.
This manuscript is interesting and well written, however, there are several major issues for acceptance.
1. In the analysis of Trim22 mRNA and protein expression, the authors examined liver tissues from others sets. In addition, the analysis of the serum Trim22 expression was also performed in other 30 LC patients and 39 HCC patients. Thus, the patients’ background should be described, such as the serum AFP levels, HBV infection, and the size of HCC, because the authors state that Trim22 is useful for diagnosis of early stage HCC with normal AFP levels. In addition, Trimm22 expression could be affected with HBV infection. To clarify this point is important in this study.
2. The size of HCC is relatively large (> 3cm: 81%) in this study, thus the diagnosis of HCC is seemed to be easy by CT or MRI. Not all patients with normal AFP is early stage of HCC, because AFP level is not elevated in 40% of HCC as authors mentioned. If the authors discuss that Trim22 is useful for early diagnosis of HCC, the patients with normal AFP and early stage (small size) of HCC should be recruited.
3. The authors select the 41 proteins were selected for MRM analysis, among the 114 proteins detected by LC-ESI-MS/MS analysis. The reason why the 41 proteins were selected is uncertain.